# Application of an Endothelial Cell Culture Assay for the Detection of Neutralizing Anti-Clostridium Perfringens Beta-Toxin Antibodies in a Porcine Vaccination Trial

**DOI:** 10.3390/toxins11040225

**Published:** 2019-04-15

**Authors:** Olivia K. Richard, Sven Springer, Jacqueline Finzel, Tobias Theuß, Marianne Wyder, Beatriz Vidondo, Horst Posthaus

**Affiliations:** 1Institute of Animal Pathology, Vetsuisse Faculty, University of Bern, 3012 Bern, Switzerland; olivia.richard@vetsuisse.unibe.ch (O.K.R.); marianne.wyder@vetsuisse.unibe.ch (M.W.); 2IDT Biologika GmbH, Business Unit Animal Health, Research and Development, 06861 Dessau-Rosslau, Germany; sven.springer@idt-biologika.de (S.S.); Jaqueline.Finzel@idt-biologika.de (J.F.); tobias.theuss@idt-biologika.de (T.T.); 3Veterinary Public Health Institute, Vetsuisse Faculty, University of Bern, 3012 Bern, Switzerland; beatriz.vidondo@gmx.ch

**Keywords:** necrotizing enteritis, *C. perfringens* type C, beta-toxin, neutralizing antibodies, cell culture assay, primary porcine endothelial cell

## Abstract

Background: Beta-toxin (CPB) is the major virulence factor of *Clostridium perfringens* type C, causing hemorrhagic enteritis in newborn pigs but also other animals and humans. Vaccines containing inactivated CPB are known to induce protective antibody titers in sow colostrum and neutralization of the CPB activity is thought to be essential for protective immunity in newborn piglets. However, no method is available to quantify the neutralizing effect of vaccine-induced antibody titers in pigs. (2) Methods: We developed a novel assay for the quantification of neutralizing anti-CPB antibodies. Sera and colostrum of sows immunized with a commercial *C. perfringens* type A and C vaccine was used to determine neutralizing effects on CPB induced cytotoxicity in endothelial cells. Antibody titers of sows and their piglets were determined and compared to results obtained by an ELISA. (3) Results: Vaccinated sows developed neutralizing antibodies against CPB in serum and colostrum. Multiparous sows developed higher serum and colostrum antibody titers after booster vaccinations than uniparous sows. The antibody titers of sows and those of their piglets correlated highly. Piglets from vaccinated sows were protected against intraperitoneal challenge with *C. perfringens* type C supernatant. (4) Conclusions: The test based on primary porcine endothelial cells quantifies neutralizing antibody activity in serum and colostrum of vaccinated sows and could be used to reduce and refine animal experimentation during vaccine development.

## 1. Introduction

*Clostridium (C.) perfringens* type C causes necrotizing enteritis (NE) in animals but also humans [1]. The main and essential virulence factor of pathogenic strains is beta-toxin (CPB), a 35 kDa toxin belonging to the beta-pore-forming toxin family [2]. The toxin is responsible for endothelial damage leading to acute and massive intestinal hemorrhage, necrosis of the small intestinal mucosa and acute death of affected hosts [3,4,5]. It has long been known that immunoprophylaxis using vaccines based on inactivated *C. perfringens* type C culture supernatants is effective in preventing the disease [6,7,8]. As the disease is primarily a problem for the porcine industry, commercially available vaccines are readily used in veterinary practice. Sows are vaccinated during every gestation period and newborn piglets are passively protected by antibodies taken up from the colostrum [9]. Although *C. perfringens* type C secretes many different toxins [10], antibodies against CPB are most likely essential for protective immunity, as the toxin is essential for disease development [11,12]. Because the pathogen can persist for several years on farms [13] continuous vaccination combined with good management and hygiene procedures is essential to maintain a low pathogen burden and to protect pig herds from the re-occurrence of the disease [14]. According to Hogh [15] gilts need to be immunized twice, at approximately days 70 and 100 of gestation, in order to induce a significant increase of anti-CPB antibodies in the colostrum. Subsequently, multiparous sows should be re-vaccinated prior to each farrowing in order to provide sufficient passive immunity to their piglets. This vaccination scheme is believed to ensure protective antibody titers [15]. Antibody titers against CPB used to be evaluated using mouse or guinea pig injection models [8,9,16,17,18], however, such methods should be replaced by in vitro assays. Currently, ELISA tests for titration of total amounts of anti-CPB antibodies in sera and colostrum can be applied [19]. These tests are mainly used during regulatory processes in vaccine development and licensing, but do not differentiate toxin neutralizing from non-neutralizing antibodies. Solanki et al. [20] recently used an in vitro neutralization assay on THP-1 cells to measure neutralizing capacity of serum from mice immunized against CPB. We previously showed that cultured primary porcine and human endothelial cells are highly sensitive to CPB, and that this toxicity can be inhibited by neutralizing anti- CPB antibodies [3,4]. Therefore, we aimed to apply a cell culture assay for the detection of neutralizing antibodies in serum and colostrum samples of pigs. The advantage of a porcine endothelial cell-based assay over human THP-1 or HUVEC cells would be that cells used are derived from the target species for which the vaccine is developed. In addition, porcine endothelial cells closely resemble the natural target cells, which have been shown to be endothelial cells in the intestinal mucosa [21]. The cell culture assay was used on serum and colostrum samples from a laboratory vaccination trial for the licensing process of a new vaccine against *C. perfringens* type C.

## 2. Results

### 2.1. Total Anti-Beta-Toxin Antibodies Determined by ELISA

For this study, 10 pregnant gilts (sows in first pregnancy) were vaccinated using a newly developed commercially available vaccine against *C. perfringens* type C (ENTEROPORC AC, IDT Biologika GmbH, Germany) according to the manufacturer’s recommendation. As a control, 10 pregnant gilts received 2.0 mL of physiological NaCl at the same time points. The scheme of vaccination, serum and colostrum sampling used is shown in Figure 1. To show the specificity of the test, another control of 3 pregnant gilts were vaccinated using a commercially available vaccine against *C. perfringens* type A (CLOSTRIPORC A, IDT Biologika GmbH, Germany), which does not contain beta-toxoid.

In the control sera and colostrum, no antibodies against CPB were detected by ELISA. Likewise no antibodies were detected in gilts of the vaccination group before the first vaccination (Figure 2a). After the first vaccination, 4 of 10 gilts showed low total antibody titers (range: 0.016–0.556 AU/mL, antibody units per mL, Figure 2a). In 9 of 10 gilt serum samples and all colostrum samples, antibody titers were detected after the second vaccination (range: 0.834–15 AU/mL, Figure 2a,b). All piglets of these gilts had detectable antibody titers, which decreased over the period of four weeks (Figure 2c). After farrowing, the antibody titers of the sows were decreased (range: 0.021–5.06 AU/mL) compared to values ante partum (a.p.) and significantly increased after the booster vaccination administered during the second pregnancy (range: 4.25–37.5 AU/mL, Figure 2a). After this booster vaccination, the colostrum antibody titers were significantly higher compared to those at the end of the first pregnancy (Figure 2b). In addition, the piglets of the second litter had significantly higher antibody titers compared to piglets of the first litter from the same sows. Similar to piglets from the first litters, piglet antibody titers decreased over the period of 4 weeks post-partum (p.p.) (Figure 2c). In 170 out of 195 samples from the non-vaccinated control sows and their piglets, no anti-CPB antibodies were detected. In 15 samples very low positivity was detected by ELISA (range: 0.008–0.096 AU/mL, Figure 2a–c). 

#### 2.1.1. Cell Culture Assay

Based on our previous findings demonstrating that primary porcine aortic endothelial cells (PAEC) are highly sensitive to recombinant CPB and that the cytotoxic effects can be eliminated by neutralizing monoclonal anti-CPB antibodies [3], we developed a cell culture based neutralization assay. In principle, upon pre-incubation of a constant amount of recombinant CPB (rCPB) with serum or colostrum samples containing neutralizing anti-CPB antibodies, the cytotoxic effect of the toxin should be inhibited. 

In a first step we tested the specificity of the cell culture assay on serum and colostrum samples of sows. 200 ng/mL rCPB were pre-incubated with serum or colostrum samples (1:2 dilution in cell culture medium), resulting in 100 ng/mL rCPB, for one hour at room temperature. PAEC were then incubated with these samples for 24 h and assessed by light microscopy (Figure 3). As positive controls, rCPB was pre-incubated with the international standard anti-CPB serum or monoclonal anti-CPB antibodies (10A2, USDA, Aimes Iowa) (Figure 3). Serum and colostrum samples of sows vaccinated against *C. perfringens* type C showed inhibitory effects on the cytotoxicity of rCPB, indicating presence of neutralizing antibodies against CPB. In contrast, serum and colostrum of sows vaccinated against *C. perfringens* type A and serum and colostrum of non-vaccinated sows did not neutralize the cytotoxic effect of rCPB on PAEC (Figure 3). 

We then determined neutralizing antibody levels of all sow and piglet samples of the vaccination trial. 200 ng/mL rCPB were pre-incubated with two-fold serial dilutions of the respective serum and colostrum samples, resulting in 100 ng/mL rCPB, followed by incubating PAEC for 24 h and determination of cell viability. Using pre-incubations of 200 ng/mL rCPB with serial two-fold dilutions of the World Health Organization International Standard Serum of equine origin (antibody titer 4770 international units per ml (IU/mL), National Institute for Biological Standards and Control, Potters Bar, Hertfordshire, EN6 3QG, UK) we determined the minimal neutralizing antibody titer at 0.58 IU/mL. The antibody titer was calculated by multiplying the minimal neutralizing antibody titer of 0.58 IU/mL with the last dilution step of serum and colostrum samples providing 100% neutralization of 100 ng/mL beta-toxin.

No neutralizing anti-beta-toxin antibodies were detected in any sample of the sows and piglets from the non-vaccinated control group (Figure 4a–c). In the group vaccinated against *C. perfringens* type C neutralizing antibodies were not detected in any serum from gilts before and after the first vaccination (time points B0 and B1, Figure 4a). After the second vaccination (B2), neutralizing antibodies were detected in the serum of 90% of the gilts (range: 2.385–19.08 IU/mL) (Figure 4a). Additionally, colostrum of gilts contained neutralizing anti-CPB antibodies in a range from 4.77 to 610.56 IU/mL (Figure 4b). The association between antibody titers of the colostrum and the serum can be described with the following regression model: Log(Y + 0.01) = 0.67 + Log(X + 0.01)*0.88, where Y is the antibody titer of the serum and X is the antibody titer of the colostrum (*p*-value < 0.01, R2 = 0.42, Spearman-rank correlation coefficient r = 0.69, *p*-value = 0.01). Neutralizing antibodies were detected in 85% of the serum samples from piglets of gilts. The titers ranged from 2.385 to 152.64 IU/mL (Figure 4c). Antibody titers subsequently decreased over 4 weeks p.p. (Figure 4c). In 15% of these piglets, no neutralizing effect was measured using the cell culture assay. In between the two pregnancies, the serum antibody titers of the sows (time point B3) significantly decreased compared to the p.p. time point (Figure 4a). In 50% of the serum samples from sows, no neutralizing effect was detected using the cell culture assay, in the other 50%, the antibody titers ranged from 2.385 to 9.54 IU/mL. Four days p.p. of the second farrowing (after booster vaccination 2 weeks a.p.) the neutralizing antibody titers were significantly increased in all sows (range: 9.54–305.28 IU/mL) (Figure 4a). Colostrum antibody titers after the second farrowing were significantly higher (range: 152.6–9768.96 IU/mL) than those from the first farrowing (Figure 4b). The colostrum antibody titer at the second farrowing highly correlated with the serum antibody titers p.p. The association between antibody titers of the colostrum and the serum can be described with the following regression model: Log(Y + 0.01) = −0.42+ Log(X + 0.01)*0.78, where X is the antibody titer of the serum and Y is the antibody titer of the colostrum (*p*-value < 0.01, R2 = 0.99, Spearman-rank correlation coefficient r = 0.96, *p*-value > 0.01, Figure 4d). All serum samples of piglets form the second litters contained neutralizing antibodies (range: 19.08–152.64 IU/mL). The mean antibody titer in the first week p.p. of these piglets was 3.6 times higher compared to piglets from the first litter (Figure 4c).

The International Units per ml measured by use of the cell culture assay strongly correlated with the AU/mL measured by ELISA with a Spearman-rank correlation coefficient r = 0.92., *p*-value < 0.01 (Figure 5). Their association can be described with the following regression model: Log (Y + 0.01) = −0.09 + Log (X + 0.01)*0.84, where X is the total antibodies in IUs measured by ELISA and Y is the neutralizing antibodies measured by the cell culture (*p*-value < 0.01, R2 = 0.8984). 

#### 2.1.2. Challenge Experiment

The results of the challenge experiments are shown in Table 1. In total 20 piglets from 10 vaccinated gilts and 16 piglets from 8 control gilts were challenged in the first trial. No piglets from vaccinated gilts exhibited serious clinical signs associated with the intoxication. In the control group 13 out of 16 (81.25%) challenged piglets showed distinct clinical signs. 11 piglets out of 16 (68.75%) had to be euthanized due to the severity of clinical signs. In the second trial 18 piglets from 9 vaccinated gilts and 20 piglets from 10 control gilts were challenged. All piglets from vaccinated gilts were protected. In the non-vaccinated control group 19 out of 20 (95.00%) challenged piglets showed distinct clinical signs. Out of 20 piglets, 18 (90.00%) had to be euthanized.

## 3. Discussion

We describe the establishment of a cell culture assay for the specific detection and quantification of neutralizing antibodies against the *C. perfringens* beta-toxin (CPB). This test system makes use of the high susceptibility of primary porcine aortic endothelial cells to CPB. As CPB is the essential virulence factor of *C. perfringens* type C [11,12], antibody mediated neutralization of the toxin can be regarded as a reliable measure for protective immunity induced by vaccines against *C. perfringens* type C. The specificity of our test was shown by detecting neutralizing anti-CPB antibodies in serum of sows vaccinated against *C. perfringens* type C, but not *C. perfringens* type A or non-vaccinated animals. Sensitivity was shown by accurately detecting low and high antibody titers, respectively. The antibody titers measured by our cell culture assay strongly correlated with those determined by ELISA. This demonstrates that both methods are suitable to detect anti-CPB antibody titers in the serum and colostrum of pigs. The cell culture assay has the advantage of demonstrating the toxin neutralizing capacity of antibodies. For *C. perfringens* type C vaccine development and licensing the European Pharmakopoe lists a mouse neutralization assay (MNT) to prove the induction of immunity, which neutralizes the toxic activity of *C. perfringens* type C supernatants [22]. Our test system offers the possibility of an alternative approach to such animal experiments. In this respect, the use of cell cultures derived from the target species of the vaccine which in addition, closely resembles those cells targeted by the toxin in vivo [21], is an advantage of the test system. Although THP-1 cells have been shown to be a suitable readout system a dose of 2.5 µg rCPB/2 × 10^4^ cells in 100 µl medium was used to achieve toxicity in these cells [20]. Compared to THP-1 cells, the CPB dose required for intoxication of primary porcine endothelial cells used in our assay was 10 ng rCPB/ 4 × 10^4^ cells in 100 µL medium. Thus, porcine endothelial cells seem to be more susceptible to rCPB and might therefore represent a more reliable readout system. Although our new assay requires the propagation of primary cell cultures, stock cultures are readily produced [3] and therefore the test can easily be performed in laboratories equipped for cell culture work. Alternatively, commercially available primary human endothelial cells, such as HUVEC, could be used as these cells are reported to be highly sensitive to CPB and the toxic effect of CPB can be inhibited by neutralizing antibodies [4,23]. These cells would be easier to obtain; however, they are not derived from the target species for which the vaccine is produced for. In addition, porcine endothelial cells might exhibit less individual variations between different donor pigs as they can be derived from pigs with more similar and uniform genetic backgrounds compared to HUVEC. Like HUVEC, primary porcine endothelial cells cannot be propagated indefinitely and from our experience should not be used after 10 subsequent passages (Wyder, personal observation). Generation of a cell line by immortalization of porcine endothelial cells, which then could be propagated more readily, might circumvent this problem. Thus, further research and development on this cell culture assay could lead to a system that can reduce, refine, or replace lethal animal testing during the process of vaccine development for use in porcine production.

Furthermore, we demonstrated the development of neutralizing anti-beta-toxin antibody titers in sows and piglets in a laboratory vaccination study, comprised of two subsequent trials in gilts and multiparous sows. The study was not designed to determine the minimal protective antibody level for oral infections in piglets, however the initial immunization of gilts generated a mean colostrum antibody titer in a range that was reported to protect piglets [15]. When sows were boostered during their second pregnancy, the mean colostrum antibody titer and the mean antibody titer in piglet sera were significantly higher at farrowing compared to the first farrowing, which supports the currently recommended vaccination scheme with one booster vaccination prior to every following farrowing [15]. Challenge experiments using an intraperitoneal application model in piglets showed that antibody titers achieved in piglets were fully protective after two and three immunizations during the first and second pregnancy. Although this assay does not determine protection against disease after oral infection, which is the natural route of infection [1], results indicate that achieved antibody levels in piglets are protective. Our assays could now be used to determine minimal protective neutralizing antibody titers to prevent disease after oral infection. However, such experiments would have to take different infectious doses into account, as we have no exact knowledge about the amount of *C. perfringens* type C ingested by piglets from the environment under natural conditions.

Taken together we demonstrate that ELISA and cell culture assays can be applied to demonstrate total but also neutralizing antibody levels against *C. perfringens* beta-toxin in serum and colostrum samples of pigs. These methods could be used to demonstrate protective antibody levels in sows and piglets in vaccination trials but also in practice to investigate herd immunity against *C. perfringens* type C enteritis.

## 4. Materials and Methods 

### 4.1. Vaccination Trial

The laboratory vaccination study, including the challenge experiments, was conducted according to the German law for Animal Welfare and registered under the animal experiment announcement Nos. IDT-A-03b-2012 and IDT-A-04b-2012. In the first trial 20 pregnant gilts (German Landrace x Duroc, 8–10 months old) were randomly assigned to two groups of 10 sows each. Group 1 (vaccination group) was immunized against *C. perfringens* type C with the newly developed *C. perfringens* type A/C toxoid vaccine ENTEROPORC AC batch 001 07 14 (IDT Biologika GmbH, Germany). The vaccine contains min. 125 relative units (rU) alpha toxoid, min. 3354 rU/mL beta toxoid and 770 rU beta2 toxoid per milliliter. For the first farrowing, sows received two separate injections of 2.0 mL of the vaccine intramuscularly (i.m.) 5 and 2 weeks a.p. Two sows developed transient mastitis and could not be included in the sampling. For a follow up study (second trial) the same sows were inseminated again and received a booster immunization with 2.0 mL of the vaccine i.m. two weeks before the second farrowing. Group 2 (non-vaccinated control group) received 2.0 mL of physiological NaCl i.m. at the same time points. Blood samples from each gilt were collected before (B0) and after the first immunization (B1) and 4 days p.p. (B2). During the second pregnancy, blood samples were taken before the booster vaccination (B3) and 4 days p.p. Colostrum was collected at birth (Col1 and Col2). In addition, blood samples were collected from two piglets per litter: one (P-B1), two (P-B2), three (P-B3), and four weeks p.p (P-B4) (Figure 1). Specificity of the cell culture assay was tested using samples of sows immunized with CLOSTRIPORC A (IDT Biologika GmbH, Germany) containing inactivated *C. perfringens* type A supernatant, including alpha- and beta2-toxoids but no beta-toxoid. The same vaccination scheme as for ENTEROPORC AC was applied. Serum and colostrum samples of three sows (B0, B1, B2, colostrum) and six piglets (P-B1) were tested. Serum and skimmed milk were generated by centrifugation of blood samples (3500× *g*, 10 min) and colostrum (23,000× *g*, 20 min). The samples were stored in aliquots at −20 °C until examination. 

### 4.2. Recombinant Beta-Toxin

The clone for expression of the recombinant beta-toxin (rCPB) was established at the Institute of Hygiene and Infectious Diseases (Justus Liebig University Giessen, Germany). For toxin production an agitated culture in LB broth was performed (37 °C, 180 rpm). Toxin production was induced at an optical density of 0.5 (OD550) by addition of 0.05 µg/mL of anhydrotetracycline followed by a further cultivation (4 h, 32 °C, 180 rpm). The cells were then separated via centrifugation (5000× *g*, 25 min, 10 °C) followed by a resuspension in physiological saline solution. Cell lysis was performed with a FRENCH-Pressure-Cell-Press (Sim-Aminco Spectronic Instruments, USA) during two passages (pressure >900 psi). The purification of the toxin was done with a StrepTactin-Matrix (Institute of Bioanalytics, Göttingen, Germany). Quantification of rCPB was performed as described by Gurtner et al. [3]. Aliquots were stored at −20 °C.

### 4.3. ELISA 

The ELISA was performed as a sandwich-ELISA. Microtiter plates (MaxiSorp^®^, Nunc GmbH, Wiesbaden, Germany) were coated with a 1:800 dilution of a monoclonal anti-CPB antibody (clone 3A6, IDT Biologika GmbH). Unspecific binding sites were blocked with milk powder supplemented phosphate buffered saline (PBS). Subsequently, the primary antibody was saturated with rCPB (>17,000 AU/mL, in house preparation). One well without addition of rCPB on every microtiter plate served as negative control for subtraction of the unspecific binding capacity of the serum. This was followed by a washing step with PBS supplemented with Tween 20 (PBST) (Merck, Germany). The sera and a standard serum (positive control; hyperimmune serum of a pig) were then added in 2-fold dilution series (incubation for 1 h at 37 °C). After a further washing step in PBST, a goat anti-pig horseradish peroxidase conjugated antibody (1:10,000, Bethyl Laboratories, USA) was added. SeramunBlau (Seramun Diagnostica GmbH, Wolzig, Germany) was used as chromogen according to the manufacturer’s instruction. The reaction was stopped with 0.5 M sulfuric acid. The plates were subsequently measured at a wavelength of 450 and 620 nm using a plate reader (Sunrise, Tecan Group Ltd., Männedorf, Switzerland). The antibody titer of the samples are given in antibody units per ml (AU/mL).

### 4.4. Cell Culture Assay

Primary porcine aortic endothelial cells (PAEC) were cultured on 96 well plates in Dulbecco’s Modified Eagle Medium (DMEM + Glutamax, Gibco^®^) containing 1% L-Glutamin, 100 U/mL penicillin, 100 u/mL streptomycin, 0.25 µg/mL Amphotericin B, and 10% fetal calf serum, at 37 °C and 5% CO_2_. 1 × 10^4^ cells/cm^2^ were seeded 5 days prior to the experiment and grown to confluency. 60µl of cell culture medium with a toxin concentration of 200 ng/mL rCPB of the same batch as used for ELISA testing were incubated with 60µl of two fold serial dilutions of serum or colostrum samples (in cell culture medium) for one hour at room temperature. PAEC were then incubated with 100 µL of these samples for 24 h at 37° with 5% CO_2_. This resulted in a final incubation of appr. 4 × 10^4^ cells with a total of 10 ng CPB in 100 µL medium (100 ng/mL).

Monoclonal mouse anti-CPB antibody 10A2 (MAb-CPB, USDA, Aimes Iowa, USA) and the *C. perfringens* beta antitoxin, 2nd International Standard (2CPBETAAT; from NIBSC, Potters Bar, Hertfordshire, England; antibody concentration of 4770 IU/mL) were used as positive controls for neutralizing antibodies. Serum and colostrum of non-vaccinated sows were used as negative controls. Cell viability was determined using a redox dye (CellTiter Blue^®^ (CTB), Promega Corporation, Madison, USA). 10µl of the CTB were added to each well and incubated for 4 h at 37° with 5% CO_2_. The proportion of viable cells was then measured as fluorescence at 560 and 590 nm using the Enspire^®^ plate reader (EnSpire^®^ Multimode Plate Reader by PerkinElmer). A blank fluorescence composed of a dead cell monolayer (cell death induced by beta-toxin), and serum or colostrum dilutions was measured for each dilution step. Fluorescence values for individual wells were determined by subtracting the blank value from the total fluorescence value of each well. The fluorescence value of each well was then correlated to the fluorescence values of the corresponding control serum or colostrum dilution. The World Health Organization International Standard Serum of equine origin (antibody titer 4770 IU/mL, National Institute for Biological Standards and Control, Potters Bar, Hertfordshire, EN6 3QG, UK) neutralized 100 ng/mL of CPB at 0.58 IU/mL. This value was multiplied with the last dilution step of serum and colostrum samples providing 100% neutralization of 100 ng/mL beta-toxin. The International Standard Serum was dissolved in PBS and diluted with completed medium. Following the fluorescence measurements, cells were fixed with 4% paraformaldehyde and stained with rapid staining dye (Hemacolor^®^). Cytopathic effects and confluency of the monolayers were evaluated by light microscopy and the percentage of intact cells was estimated independently by two investigators.

### 4.5. Challenge Experiments

For the production of the challenge material, *C. perfringens* type C strain No. MB-yellow-130 was cultivated in TVS medium. After separation of the biomass by centrifugation (10,000× *g*), the supernatant was concentrated by ultrafiltration and ammonium sulfate precipitation. The pellet was resuspended in PBS, sterile filtrated, freeze dried and stored at 2 to 8 °C. 

For the challenge experiment, one vial of the freeze-dried toxin was dissolved in 19.0 mL sterile TVS medium. Two piglets from each of the vaccinated and control sows at the 1st and 2nd farrowing were challenged by intraperitoneal (i.p.) injection at the 2nd day of life with 2.0 mL of the dissolved toxin. After intoxication the clinical parameters general condition, posture, behavior, milk uptake, nutritional status of the piglets, respiration, ascites, skin turgor, and diarrhea were assessed until the end of the suckling period (Table 2). While assessing the clinical parameters an animal was deemed to be sick when: (i) it had a summative clinical score of 5 or higher; (ii) it developed ascites; (iii) it lagged behind its littermates in weight development due to the symptoms triggered by the intoxication (runts). The piglets were euthanized according to animal welfare reasons if the following occasions were present: (i) acute onset of clinical disease: disturbed general condition combined with a somnolence, lack of milk uptake and/or a failure of weight gain; (ii) chronic progression of clinical disease: severe ascites and/or an impaired breathing in combination with a stagnation or rather a decrease in body weight. At the end of the trial, the number of diseased and euthanized piglets in the vaccinated and control group were evaluated and compared using Fisher’s Exact Test. In accordance with monograph 04/2013/0360 (Ph. Eur.) protection was deemed to be shown if ≤20% of the piglets of the vaccinated sows and ≥80% of the piglets of the control sows were considered sick or had to be euthanized after intoxication.

### 4.6. Statistics

Statistics were done using the NCSS software (Nashville, USA, http://www.ncss.com) and SPSS (SPSS Inc.). Shapiro–Wilk test was used to test normality. As the antibody titers were not normally distributed, the logarithm transformation (log(y + 0.01)) was used. The differences in antibody titers between the treatment groups were tested by means of a repeated measures analysis of variance, taking the animal identification number as a subject variable. The post hoc test used for multiple comparisons was the Tukey–Kramer Multiple-Comparison Test. The level of significance was set to 0.05. Besides, the associations between the antibody titers in serum and colostrum and measured by ELISA and the cell culture assay was evaluated by means of linear regression of the log transformed variables and Spearman–Rank correlation coefficients. The differences between the treatment groups in the challenge experiment were tested using Fisher’s Exact Test. 

### 4.7. Ethics Approval 

This study was conducted according to the German law for Animal Welfare and approved on 25 March 2012 under the animal experiment announcement Nos. IDT-A-03b-2012 and IDT-A-04b-2012.

## Figures and Tables

**Figure 1 toxins-11-00225-f001:**
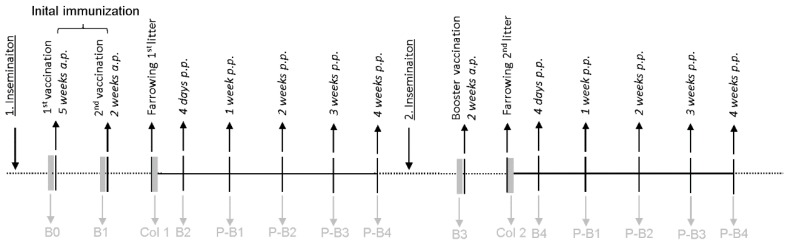
Immunization and sampling schedule for vaccination trial. Timeline of sample collection in the vaccination trial; B0: before the initial immunization, B1: after the first immunization, B2: after the second immunization, B3: before the booster vaccination during 2nd pregnancy, B4: after the booster vaccination during 2nd pregnancy, P-B1: one week p.p., P-B2: 2 weeks p.p., P-B3: 3 weeks p.p., P-B4: 4 weeks p.p.; B: blood sample of the sow, Col: colostrum, P-B: blood samples of piglets.

**Figure 2 toxins-11-00225-f002:**
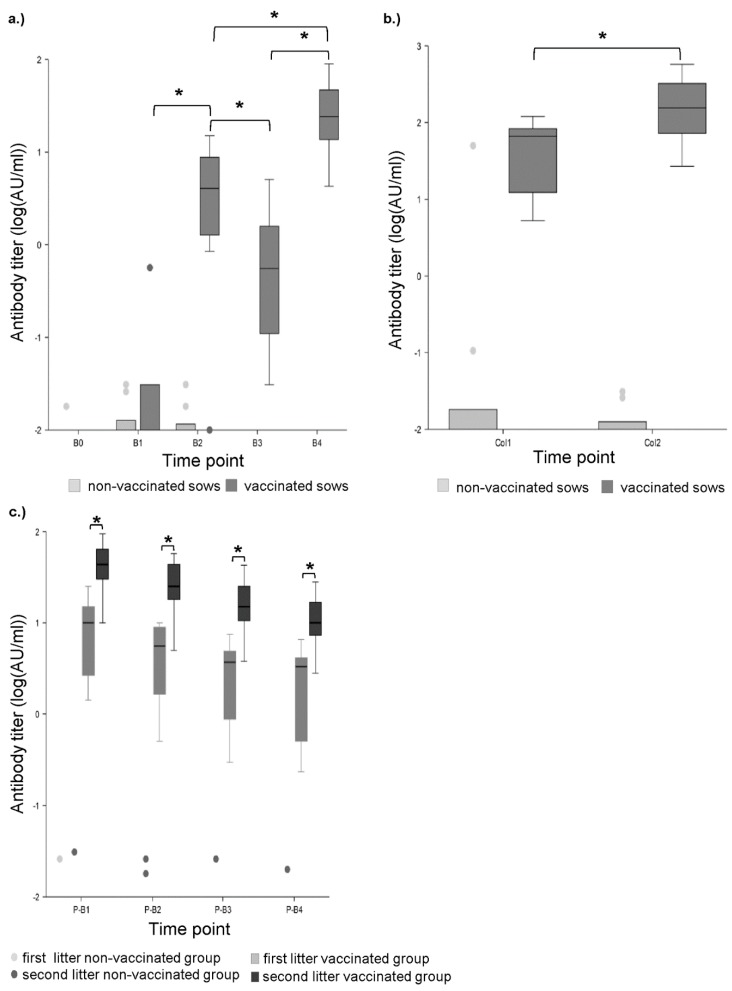
Total anti-beta-toxin (CPB) antibodies determined by ELISA. (**a**) Progression of antibody titers in sow serums measured by ELISA. B0: before the initial immunization, B1: after first immunization, B2: after second immunization, B3: before booster vaccination during 2nd pregnancy, B4: after booster vaccination during 2nd pregnancy. Asterisks indicate significant differences between groups (repeated measures ANOVA, Tukey–Kramer Multiple-Comparison test). Antibody titers significantly increased from B1 to B2 (*p* < 0.01), decreased from B2 to B3 (*p* < 0.01) and increased again after the booster immunization from B3 to B4 (*p* < 0.01). Antibody titers after the second pregnancy (B4) were significantly higher than after the first (B2) (*p* < 0.01). (**b**) Colostrum antibody titers after first (1) and second (2) farrowing measured by ELISA. Titers after the second farrowing were significantly higher (asterisks) compared to the first farrowing (*p* < 0.05) (repeated measures ANOVA, Tukey–Kramer Multiple-Comparison Test). (**c**) Progression of antibody titers in piglet serums 1 (P-B1), 2 (P-B2), 3 (P-B3), and 4 weeks p.p. (P-B4) measured by ELISA. Asterisks indicate significant differences between groups (repeated measures ANOVA, Tukey–Kramer Multiple-Comparison test). Mean antibody titers of the second litter were significantly higher compared to the first litter in the vaccinated groups (*p* < 0.01 for all time points). No neutralizing antibodies were detected in the non-vaccinated group.

**Figure 3 toxins-11-00225-f003:**
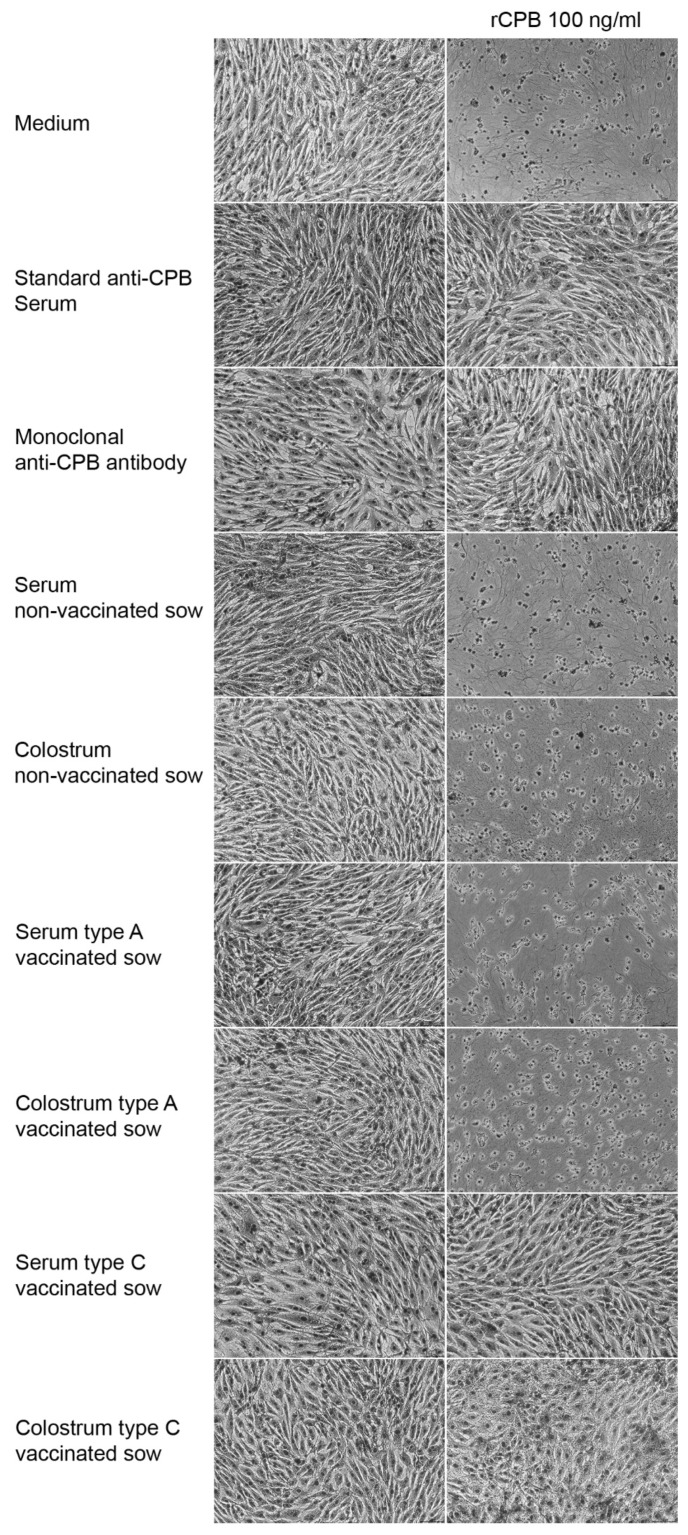
Inhibitory effect on recombinant CPB (rCPB) mediated cytotoxicity in serum and colostrum of sows vaccinated against *C. perfringens* type C. Left lane: Confluent porcine aortic endothelial cells (PAEC) were incubated for 24 h with cell culture medium containing no additives (medium), control anti-CPB antibodies, or serum (1:2 volume dilutions) and colostrum (1:2 volume dilutions) samples of sows treated as indicated. The right lane shows representative results of pre-incubation of 100 ng/mL rCPB with the same media and additives. Absence of cytopathic effects after pre-incubation with respective samples indicated presence of specific neutralizing anti-CPB antibodies in serum and colostrum samples of only those sows vaccinated against *C. perfringens* type C. Cells were fixed in paraformaldehyde, stained with Hemacolor^®^ and photographed under a light microscope (magnification 100x).

**Figure 4 toxins-11-00225-f004:**
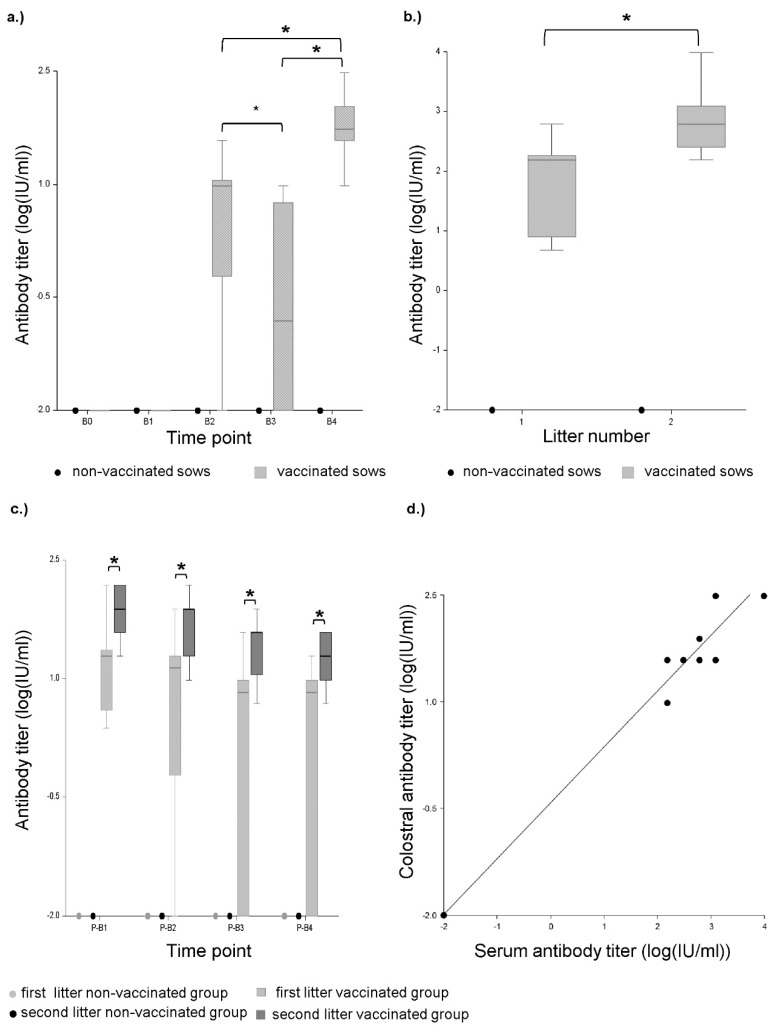
Neutralizing anti-CPB antibody titers determined by cell culture assay. (**a**) Progression of neutralizing antibody titers determined by cell culture assay in sow serums. B0: before the initial immunization, B1: after first immunization, B2: after second immunization, B3: before booster vaccination during 2nd pregnancy, B4: after booster vaccination during 2nd pregnancy. Asterisks indicate significant differences between groups (repeated measures ANOVA, Tukey–Kramer Multiple-Comparison test). Antibody titers significantly increased from B1 to B2 (*p* < 0.01), decreased from B2 to B3 (*p* < 0.05) and increased again after the booster immunization from B3 to B4 (*p* < 0.01). Antibody titers after the second pregnancy (B4) were significantly higher than after the first (B2) (*p* < 0.01). (**b**) Neutralizing colostral antibody titers after first (1) and second (2) farrowing determined by cell culture assay. Titers after the second farrowing were significantly higher (asterisks) compared to the first farrowing (*p* < 0.05) (repeated measures ANOVA, Tukey–Kramer Multiple-Comparison Test). (**c**) Progression of neutralizing antibody titers in piglet serums one (P-B1), two (P-B2), three (P-B3) and four weeks p.p. (P-B4) determined by cell culture assay. Asterisks indicate significant differences between groups (repeated measures ANOVA, Tukey–Kramer Multiple-Comparison test). Mean antibody titers of the second litter were significantly higher compared to the first litter in the vaccinated groups (*p* < 0.01 for all time points). No neutralizing antibodies were detected in the non-vaccinated group. (**d**) Scatterplot and linear regression model (continuous line, equation in results) of neutralizing antibody titers in vaccinated sows in serum *p.p*. of the second litter and colostrum determined by cell culture assay. Units: Log(IU/mL + 0.01). Each dot represents samples from one litter. One sow did not gain antibodies (value at lower left corner).

**Figure 5 toxins-11-00225-f005:**
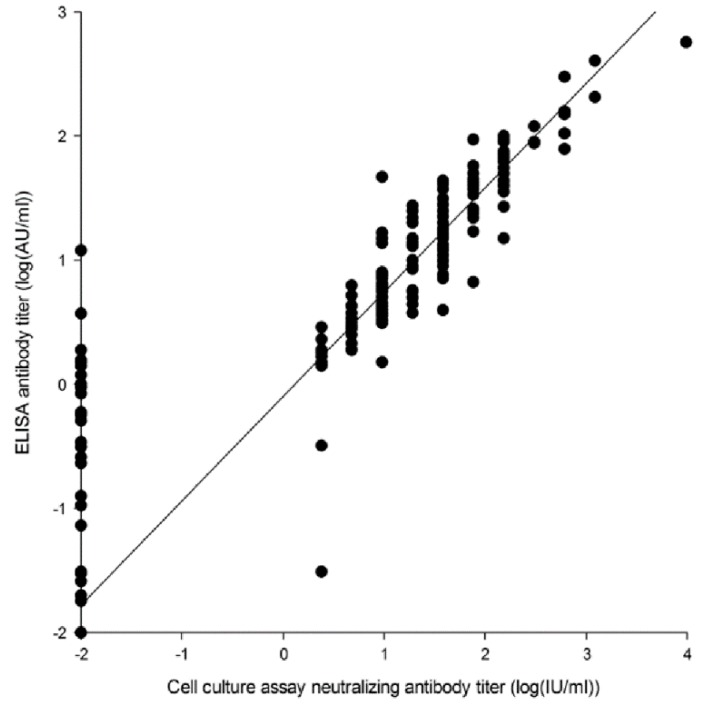
Scatterplot and linear regression model (continuous line, equation in results) of total and neutralizing antibody titers in serum and colostrum measured by ELISA and cell culture in all vaccinated sows. Total antibody titer measured by ELISA in AU/mL and neutralizing antibody titer measured by cell culture assay in Units: Log (AU/mL + 0.01) and log(IU/mL + 0.01). Each dot represents an individual sample.

**Table 1 toxins-11-00225-t001:** Results of the laboratory vaccination and challenge trials.

				Morbidity Rate		Mortality Rate	
Trial	Group	Sows (n)	Piglets (n)	n	In %	n	In %
1	2 x vaccination	10	20	0 *	0	0 *	0
1	Control	8	16	13	81.25	11	68.75
2	3 x vaccination	9	18	0 *	0	0 *	0
2	Control	10	20	19	95.00	18	90.00

* Significant differences between the vaccinated group and control group (Fisher’s exact test, level of significance *p* < 0.05 (Fisher’s exact test).

**Table 2 toxins-11-00225-t002:** Examination parameters of the piglets following intoxication.

Parameter	Clinical Findings and Score
General condition (general clinical impression)	0 = undisturbed
	1 = disturbed
Posture	0 = normal, body supported equally on all 4 limbs, animal moves physiologically
	1 = unstable gait, kyphosis
	2 = animal lies permanently
Behaviour	0 = normal, animal is alert
	1 = apathetic
	2 = somnolent
Milk up-take	0 = yes
	1 = no
Nutritional status *	0 = good
	1 = medium (spine and pelvic bone easily recognisable)
	2 = cachexia
Respiration	0 = normal
	1 = increased respiratory rate
	2 = distinct abdominal breathing
Ascites	0 = no
	1 = slight
	2 = severe
Skin turgor	0 = < 1 s (after pulling skin fold in neck region)
	1 = 1 – 3 s
	2 = > 3 s
Diarrhoea	0 = no, formed faeces
	1 = yes, pasty
	2 = yes, runny
	3 = yes, pasty or runny, bloody, or black feces

* Piglets were weighed twice weekly.

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
