# Peer review of "Application of an Endothelial Cell Culture Assay for the Detection of Neutralizing Anti-Clostridium Perfringens Beta-Toxin Antibodies in a Porcine Vaccination Trial"

_toxins, 2019, doi:10.3390/toxins11040225_

Round 1

Reviewer 1 Report

The manuscript „ Development of a cell culture assay for the detection of neutralizing anti-Clostridium 2 perfringens beta-toxin antibodies in serum and colostrum of pigs” deals with a refinement assay for measuring successful vaccination of sows and neutralizing antibody titers in sows and piglets. The manuscript is sound, and data are analyzed using appropriate statistics. The study shows clear results concerning vaccination and resulting neutralizing antibody titers against C. perfringens beta-toxin, which is the major, if not essential, pathogenicity factor of C. perfringens type C.

There are three major points that have to be addressed in a revised version of the manuscript:

 1) The authors nicely correlated ELISA and neutralizing capacity of sera in cell culture assay. What is more important is the relationship of neutralizing capacity in cell culture assay and protection of animals against C. perfringens. The authors should try to analyze data to evaluate titers that are predictive for successful vaccination.

2) This is not the first cell culture based assay for detection of neutralizing antisera from vaccinated animals. Solanki et al (PMID: 28523396) used THP-1 cells for a similar study in 2017. The authors should discuss findings and advantages of the two systems.

3) A clear statement for conflict of interest has to be included since a newly developed commercial vaccine was used for the study (ENTEROPORC AC, IDT Biologika 61 GmbH, Germany). The vaccine enteroporc is not listed on website of the company, yet. Furthermore, the used vaccine contains alpha-toxoid and beta2-toxoid from C. perfringens Type A besides beta1 toxoid from typeC. This has to be discussed, especially in relation with Fig. 4. Toxoid content should be stated in Material section

Minor comments:

There are several typos, i.e. missing blanks in concentrations or subscribed numbers.

Author Response

Thank you for reviewing our mansucript and your valuable comments. We hope we could respond adequately to them.

Reviewers comment 1:

The authors nicely correlated ELISA and neutralizing capacity of sera in cell culture assay. What is more important is the relationship of neutralizing capacity in cell culture assay and protection of animals against C. perfringens. The authors should try to analyze data to evaluate titers that are predictive for successful vaccination.

Authors response:

We agree that it would be interesting to determine the protective antibody levels in piglets for the natural disease. However, this would require extensive lethal oral inoculation trials which were far beyond the scope of our study, which was designed to evaluate the suitability of a porcine endothelial cell culture system for detection of anti-CPB neutralizing antibodies. We also are not aware of any scientific publication that adequately covers this question. As a response to this valid request we included this issue in our discussion and added a few more references of rather old studies which dealt with protective antibody levels. 

Revised paragraph in discussion, p 11, lines 267-282:

Furthermore, we demonstrated the development of neutralizing anti-beta-toxin antibody titers in sows and piglets in a laboratory vaccination study, comprised of two subsequent trials in gilts and multiparous sows. The study was not designed to determine the minimal protective antibody level for oral infections in piglets, however the initial immunization of gilts generated a mean colostrum antibody titer in a range that was reported to protect piglets [15]. When sows were boostered during their second pregnancy, the mean colostrum antibody titer and the mean antibody titer in piglet sera were significantly higher at farrowing compared to the first farrowing, which supports the currently recommended vaccination scheme with one booster vaccination prior to every followingfarrowing [15]. Challenge experiments using an intraperitoneal application model in piglets showed that antibody titers achieved in piglets were fully protective after two and three immunizations during the first and second pregnancy. Although this assay does not determine protection against disease after oral infection, which is the natural route of infection [1], results indicate that achieved antibody levels in piglets are protective. Our assays could now be used to determine minimal protective neutralizing antibody titers to prevent disease after oral infection. However, such experiments would have to take different infectious doses into account, as we have no exact knowledge about the amount of C. perfringens type C ingested by piglets from the environment under natural conditions.

Reviewers comment 2:

This is not the first cell culture based assay for detection of neutralizing antisera from vaccinated animals. Solanki et al (PMID: 28523396) used THP-1 cells for a similar study in 2017. The authors should discuss findings and advantages of the two systems.

Authors response:

Thank you for this comment. We were not aware that this test was included in this manuscript. We have included this issue in the introduction and discussion. To compare the toxin doses used in the two studies we added more detail to our materials and methods on the cell culture system. We also corrected a typo mistake leading to incorrect information about cell numbers seeded into 96 wells.

Revised paragraph in Introduction, p 2, lines 47-62:

Antibody titers against CPB used to be evaluated using mouse or guinea pig injection models [8,9,16-18], however, such methods should be replaced by in vitro assays. Currently, ELISA tests for titration of total amounts of anti-CPB antibodies in sera and colostrum can be applied [19]. These tests are mainly used during regulatory processes in vaccine development and licensing, but do not differentiate toxin neutralizing from non-neutralizing antibodies. Solanki et al [20] recently used an in vitro neutralization assay on THP-1 cells to measure neutralizing capacity of serum from mice immunized against CPB. We previously showed that cultured primary porcine endothelial cells are highly sensitive to CPB, and that this toxicity can be inhibited by neutralizing anti- CPB antibodies [3].Therefore, we aimed to develop a cell culture assay for the detection of neutralizing antibodies in serum and colostrum samples of pigs. The advantage of a porcine endothelial cell-based assay over THP-1 cell would be that cells used are derived from the target species for which the vaccine is developed and that porcine endothelial cells more closely resemble the natural target cells, which have been shown to be endothelial cells in the intestinal mucosa [21]. For the establishment of the cell culture assay, serum and colostrum samples from a laboratory vaccination trial for the licensing process of a new vaccine against C. perfringens type C were used.

Revised paragraph in discussion, p 11, lines 252-259:

In this respect, the use of cell cultures derived from the target species of the vaccine which in addition, closely resembles those cells targeted by the toxin in vivo [21], is an advantage of the test system. Although THP-1 cells have been shown to be a suitable readout system a dose of 2.5 µg, rCPB/2 x 104 cells in 100 µl medium was used to achieve toxicity in these cells [20]. Compared to THP-1 cells, the CPB dose required for intoxication of primary porcine endothelial cells used in our assay was 10 ng rCPB/ 4 x 104 cells in 100 µl medium. Thus, porcine endothelial cells seem to be more susceptible to rCPB and might therefore represent a more reliable readout system.  

Revised materials and methods section, p 13, Lines 346-354:

4.4. Cell culture assay

Primary porcine aortic endothelial cells (PAEC) were cultured on 96 well plates in Dulbecco's Modified Eagle Medium (DMEM + Glutamax, Gibco®) containing 1% L-Glutamin, 1 % Antibiotics/Antimycotics and 10% fetal calf serum, at 37°C and 5% CO2. 1x104 cells/cm2 were seeded 5 days prior to the experiment and grown to confluency. 60µl of cell culture medium with a toxin concentration of 200ng/ml rCPB of the same batch as used for ELISA testing were incubated with 60µl of two fold serial dilutions of serum or colostrum samples (in cell culture medium) for one hour at room temperature. PAEC were then incubated with 100 µl of these samples for 24 hours at 37° with 5% CO2. This resulted in a final incubation of appr. 4x104 cells with a total of 10 ng CPB in 100 µl medium (100 ng/ml).

Reviewers comment 3:

A clear statement for conflict of interest has to be included since a newly developed commercial vaccine was used for the study (ENTEROPORC AC, IDT Biologika 61 GmbH, Germany). The vaccine enteroporc is not listed on website of the company, yet. Furthermore, the used vaccine contains alpha-toxoid and beta2-toxoid from C. perfringens Type A besides beta1 toxoid from typeC. This has to be discussed, especially in relation with Fig. 4. Toxoid content should be stated in Material section

Authors responses:

3.1. The toxoid contents were added in the chapter “material and methods”:

P 12, lines 294-298:

Group 1 (vaccination group) was immunized against C. perfringens type C with the newly developed C. perfringens type A/C toxoid vaccine ENTEROPORC AC batch 001 07 14 (IDT Biologika GmbH, Germany). The vaccine contains min. 125 relative units (rU) alpha toxoid, min. 3354 rU/ml beta toxoid and 770 rU beta2 toxoid per milliliter.

3.2. The vaccine Enteroporc AC was designed for commercial use by IDT Biologika GmbH and is currently registered under the DCP reference number for application DE/V/0271/001.

We add this statement to the conflict of interest statement on the webpage of Toxins. Other than that we are not aware which additional statement should be mentioned under conflict of interest. We would of course be willing to follow additional guidance by the reviewer/editor.

3.3. For production-technical reasons the market introduction of the vaccine is planned in the 1st quarter 2020. That is the reason why the vaccine is not listed on the website yet.

3.4. The objective of this collaborative study between our institutions was the proof of the cell culture assay for the detection of neutralizing anti-Clostridium perfringens beta-toxin antibodies in serum and colostrum. The detection of antibodies against alpha- and beta2 toxins of C. perfringens type A and the relevance of the antibodies for the efficacy against C. perfringens type A will be addressed in a further study and discussed as part of another publication by the IDT co-authors of this study.

Reviewers comment 4:

There are several typos, i.e. missing blanks in concentrations or subscribed numbers.

We corrected these mistakes in the manuscript. They are visible as tracked changes.

Reviewer 2 Report

The manuscript entitled “Development of a cell culture assay for the detection of neutralizing anti-Clostridium perfringens beta-toxin antibodies in serum and colostrum of pigs” reports a cell-based method to quantify the neutralizing antibody activity in serum and colostrum of vaccinated pigs. The cell-based measurements of neutralizing antibodies showed a significant correlation with the measurement observed from ELISA method. The authors also demonstrated that the antibody titers achieved in piglets were active and protective in the challenge experiments. The study is interesting, and the paper is presented systematically. The reviewer feels the manuscript is appropriate for publication in the journal Toxins.

Author Response

Thank you for reviewing our mansucript.

We have responded to the comments of reviewer 1 in a separate letter.

Round 2

Reviewer 1 Report

The revised version of the manuscript “Development of a cell culture assay for the detection of neutralizing anti-Clostridium perfringens beta-toxin antibodies in serum and colostrum of pigs” adequately addresses two major points but still shows shortcomings regarding the central topic of developing a cell culture assay as stated in title of manuscript.

Major point:

The manuscript in its revised version still is a detailed description and validation of a vaccination trial. The authors should change title accordingly, if this was intended. The manuscript does not adequately describe development or evaluation of a cell culture assay with pro and cons. Although the revised version now refers to Solanki et al, the authors missed (?!) the opportunity to addionally refer to a former cell culture assay for neutralizing effects by specific CPB antibody using HUVECs (reference #4, Popescu et al., 2011). Isn’t a commercially available primary cell line more compliant than primary cells isolated from narcotized pigs? Again, if the manuscript aims at introducing a cell culture assay, the authors should address this properly.

Minor point:

In abstract, line 11, it should be stated that a "C perfringens type A and C vaccine" was used unless the authors can exclude supporting cross reactivity of type A vaccine for type C neutralization.

Author Response

Thank you for your comments.

Below is our point by point Response:

Major point:

The manuscript in its revised version still is a detailed description and validation of a vaccination trial. The authors should change title accordingly, if this was intended. The manuscript does not adequately describe development or evaluation of a cell culture assay with pro and cons. Although the revised version now refers to Solanki et al, the authors missed (?!) the opportunity to addionally refer to a former cell culture assay for neutralizing effects by specific CPB antibody using HUVECs (reference #4, Popescu et al., 2011). Isn’t a commercially available primary cell line more compliant than primary cells isolated from narcotized pigs? Again, if the manuscript aims at introducing a cell culture assay, the authors should address this properly.

Authors Response:

Title was changed into:

Application of an endothelial cell culture assay for the detection of neutralizing anti-Clostridium perfringens beta-toxin antibodies in a porcine vaccination trial

Abstract was modified:

(4) Conclusions: The test based on primary porcine endothelial cells quantifies neutralizing antibody activity in serum and colostrum of vaccinated sows and could be used to reduce and refine animal experimentation during vaccine development.

Last paragraph of introduction was modified and Reference Popescu et al was included (Reference nr. 4): P 2, lines 57 ff

We previously showed that cultured primary porcine and human endothelial cells are highly sensitive to CPB, and that this toxicity can be inhibited by neutralizing anti- CPB antibodies [3,4]. Therefore, we aimed to apply a cell culture assay for the detection of neutralizing antibodies in serum and colostrum samples of pigs. The advantage of a porcine endothelial cell-based assay over human THP-1 or HUVEC cells would be that cells used are derived from the target species for which the vaccine is developed. In addition, porcine endothelial cells closely resemble the natural target cells, which have been shown to be endothelial cells in the intestinal mucosa [21]. The cell culture assay was used on serum and colostrum samples from a laboratory vaccination trial for the licensing process of a new vaccine against C. perfringens type C.

Discussion was modified (p 11, lines 263-274)

Alternatively, commercially available primary human endothelial cells, such as HUVEC, could be used as these cells are reported to be highly sensitive to CPB and the toxic effect of CPB can be inhibited by neutralizing antibodies [4, 23]. These cells would be more easy to obtain, however they are not derived from the target species for which the vaccine is produced for. In addition, porcine endothelial cells might exhibit less individual variations between different donor pigs as they can be derived from pigs with more similar and uniform genetic backgrounds compared to HUVEC. Like HUVEC, primary porcine endothelial cells cannot be propagated indefinitely and from our experience should not be used after 10 subsequent passages (Wyder, personal observation). Generation of a cell line by immortalization of porcine endothelial cells, which then could be propagated more readily, might circumvent this problem. Thus, further research and development on this cell culture assay could lead to a system that can reduce, refine or replace lethal animal testing during the process of vaccine development for use in porcine production.

Minor point:

In abstract, line 11, it should be stated that a "C perfringens type A and C vaccine" was used unless the authors can exclude supporting cross reactivity of type A vaccine for type C neutralization.

Sentence changed to: Sera and colostrum of sows immunized with a commercial C. perfringens type A and C vaccine was used to determine neutralizing effects on CPB induced cytotoxicity in endothelial cells.

Round 3

Reviewer 1 Report

The authors have addressed all questions adequately. The title now fits to the Body of the manuscript. There is still one minor point that has to be taken care of:

in Para 4.4 Cell Culture Assay, first sentence, line 356/357 "1% Antibiotics/Antimycotics" has to be specified.   

Author Response

Thanks,

the sentence was changed to:

Primary porcine aortic endothelial cells (PAEC) were cultured on 96 well plates in Dulbecco's Modified Eagle Medium (DMEM + Glutamax, Gibco®) containing 1% L-Glutamin,  100U/ml penicillin, 100u/ml streptomycin, 0.25 µg/ml Amphotericin B, and 10% fetal calf serum, at 37°C and 5% CO2.